# The Role of Liquid Biopsy in Hepatocellular Carcinoma Prognostication

**DOI:** 10.3390/cancers13040659

**Published:** 2021-02-06

**Authors:** Ismail Labgaa, Augusto Villanueva, Olivier Dormond, Nicolas Demartines, Emmanuel Melloul

**Affiliations:** 1Department of Visceral Surgery, Lausanne University Hospital (CHUV), University of Lausanne (UNIL), CH-1011 Lausanne, Switzerland; ismail.labgaa@chuv.ch (I.L.); olivier.dormond@chuv.ch (O.D.); emmanuel.melloul@chuv.ch (E.M.); 2Division of Liver Diseases, Liver Cancer Program, Tisch Cancer Institute, Icahn School of Medicine at Mount Sinai, New York, NY 10029, USA; augusto.villanueva@mssm.edu; 3Division of Hematology/Oncology, Department of Medicine, Icahn School of Medicine at Mount Sinai, New York, NY 10029, USA

**Keywords:** liver cancer, biomarkers, precision medicine, circulating, prognosis

## Abstract

**Simple Summary:**

Hepatocellular carcinoma (HCC) is one of the deadliest cancer. Clinical guidelines for the management of HCC endorse algorithms deriving from clinical variables whose performances to prognosticate HCC is limited. Liquid biopsy is the molecular analysis of tumor by-products released into the bloodstream. It offers minimally-invasive access to circulating analytes like DNA, RNA, exosomes and cells. This technology demonstrated promising results for various applications in cancers, including prognostication. This review aimed to provide a comprehensive overview of the contribution of liquid biopsy in HCC prognostication. The results suggested that liquid biopsy may be a polyvalent and valuable tool to prognosticate HCC.

**Abstract:**

Showing a steadily increasing cancer-related mortality, the epidemiological evolution of hepatocellular carcinoma (HCC) is concerning. Numerous strategies have attempted to prognosticate HCC but their performance is modest; this is partially due to the heterogeneous biology of this cancer. Current clinical guidelines endorse classifications and scores that use clinical variables, such as the Barcelona Clinic Liver Cancer (BCLC) classification. These algorithms are unlikely to fully recapitulate the genomic complexity of HCC. Integrating molecular readouts on a patient-basis, following a precision-medicine perspective, might be an option to refine prognostic systems. The limited access to HCC tissue samples is an important limitation to these approaches but it could be partially circumvented by using liquid biopsy. This concept consists of the molecular analysis of products derived from a solid tumor and released into biological fluids, mostly into the bloodstream. It offers an easy and minimally-invasive access to DNA, RNA, extracellular vesicles and cells that can be analyzed with next-generation sequencing (NGS) technologies. This review aims to investigate the potential contributions of liquid biopsy in HCC prognostication. The results identified prognostic values for each of the components of liquid biopsy, suggesting that this technology may help refine HCC prognostication.

## 1. Background

Hepatocellular carcinoma (HCC) shows a worrisome epidemiological trajectory [1]. The WHO predicts over 1 million HCC-related deaths in 2030 [2]. A particular feature of HCC is that it typically arises on a chronically damaged organ, mostly cirrhosis, for which viral hepatitis, alcohol use disorder and NAFLD are common causes. This facilitates the identification of patients at risk and has enabled successful surveillance programs for early cancer detection. However, having two potentially life-threatening diseases in the same patient (i.e., HCC and cirrhosis) complicates its clinical management and prognosis prediction. Numerous attempts have been pursued to reliably prognosticate HCC, using various strategies. The most widely used classifications, like the Barcelona Clinic Liver Cancer (BCLC) algorithm [3,4], relies on clinical variables. Other approaches investigate the contribution of tumor markers like α-fetoprotein (AFP) [5] and molecular markers derived from the tumor or adjacent non-tumor samples [6,7,8,9]. Regardless of the strategy, the prognostic performance of these algorithms needs to be improved. Following a ‘precision medicine’ perspective may be a way of improving HCC prognostication. This implies access to genomic data on a patient-basis, which requires biopsy or surgical specimens for tissue samples of the tumor. This is particularly difficult in HCC for which, unlike most solid tumors, diagnosis mostly relies on imaging and tissues samples are rarely available [10,11]. In addition, tissue biopsies are associated with potential complications and should not be sequentially repeated [12].

This can be circumvented using liquid biopsy, which refers to the molecular analysis of tumor components released from a solid tumor into biological fluids like blood. These analytes include circulating tumor nucleic acids (DNA and RNAs), cells (CTCs) and exosomes (Figure 1). This technology has shown promising results for various applications in the management of different types of cancers including HCC [13,14,15]: early diagnosis [16,17], detection of minimal residual disease [18], decision-making for systemic therapies [19,20] or even to decipher complex biological traits of cancers [21,22,23,24]. This technology offers a valuable alternative to standard biopsy. Tissue biopsy is indeed invasive and associated with potential risks such as pain, bleeding or even seeding of the cancer (PMID: 18669577). Conversely to standard biopsy, liquid biopsy displays the advantages of being easily repeatable and can thereby help for monitoring, providing a dynamic picture of the disease course. In addition, it may reflect different regions of the tumor and thus recapitulate eventual intra-tumoral heterogeneity (ITH) (Figure 2) [25].

This study aims to provide a comprehensive overview of the potential contributions of each component of liquid biopsy (i.e., DNA, RNAs and cells) to HCC prognostication.

### 1.1. Circulating Tumor DNA (ctDNA)

DNA fragments are released from solid tumors into the bloodstream via active and passive mechanisms. The latter seems to predominate, being partially driven by cell necrosis and apoptosis [26]. This occurs at any tumor stage and offers minimally-invasive access to key molecular information of the tumor including genomic (copy number variations (CNV) or point mutations) as well as epigenetic (DNA methylations changes) data. Numerous studies show the value of ctDNA as a polyvalent biomarker in cancer. For example, ctDNA allowed detection of minimal residual disease (MRD) in a prospective cohort of 230 patients undergoing surgical resection of stage II colon cancer. Postoperative detection of ctDNA outperformed prognostic factors such as TNM stage, for the prediction of recurrence-free survival [18].

In HCC, a pilot study demonstrated that detection of mutations in the plasma of HCC patients was feasible and recapitulated the ones detected in tumor tissue [27]. Table 1 summarizes reports investigating ctDNA in HCC prognostication.

#### 1.1.1. Copy Number Variations (CNVs)

CNVs are potential drivers of hepatocarcinogenesis; they predominantly affect chromosomes (chr) 8 and 11 [45]. A pioneer study took advantage of this knowledge and of a mathematical model to infer CNVs using ctDNA in HCC and non-HCC patients [26]. First, it determined the typical size of DNA fragments, approximating 160 bp, thereby suggesting that DNA fragments are foremost released by apoptotic cancer cells. Second, the model performed well in distinguishing patients with chronic HBV with and without HCC with an area under the curve (AUC) of 0.93.

In plasma, a recent study targeted both CNVs and mutations using ctDNA in 34 HCC patients undergoing surgery. CtDNA was reported as a prognostic factor for survival, and it was also able to detect MRD [28]. Another study targeted VEGFA amplification in circulating-free DNA (cfDNA), assuming that it would predict response to sorafenib. Although a high concentration of cfDNA was associated with lower survival, the VEGFA ratio was not a predictor of response to therapy [29].

#### 1.1.2. Mutations

The genomic complexity of HCC–characterized by a wide spectrum of different potential driver mutations [46,47]–and the low amount of tumor DNA among the pool of cell free DNA are two major challenges when conducting mutation calling from the plasma of HCC patients. Various techniques can be applied to detect plasma mutations: droplet digital PCR (ddPCR) is better suited to targeting a small number of genes, whereas targeted-sequencing allows investigation of a larger panel of candidates. Although whole-genome sequencing is feasible, it is associated with relatively low coverage, making the interpretation of mutations calling cumbersome.

Several studies focused on *TERT promoter*, *TP53* and *CTNNB1*, as they are commonly mutated genes in HCC. In a cohort of 41 HCC patients and 10 controls, detection of these mutations was associated with shorter recurrence-free survival after liver resection [31]. This was confirmed by other studies identifying *TERT prom* and *TP53* mutations as prognostic factors of poor survival [33,34,36,37]. Targeted-sequencing of various panels of genes further confirmed the prognostic impact of ctDNA detection, associated with worse survival or higher recurrence [30]. Providing more detailed analyses than just the simple presence/absence of ctDNA, Kim et al. showed that *MLH1* mutation was specifically associated with lower survival [35], whereas von Felden et al. recently demonstrated that mutations of genes from the PI3K/mTOR pathway were predictors of non-response to tyrosine kinase inhibitors (TKI) in patients with advanced HCC [38].

Although most reports used blood samples, liquid biopsy is applicable to other types of biological fluids like urine or saliva. This was illustrated by a study demonstrating the feasibility of detecting mutations in urine samples of HCC patients. Moreover, detection of mutation preceded tumor recurrence as detected by magnetic resonance imaging (MRI) [32].

#### 1.1.3. DNA Methylation Changes

The carcinogenic role of epigenetic events like DNA methylation is well known in HCC [48,49]. Changes in DNA methylation can also be detected in cfDNA. Similar to mutations, focus can be either on specific candidate or interrogate multiple CpG sites. Several studies identified methylation changes of specific genes associated with HCC outcomes: hypomethylation of LINE-1 [42] and methylation of IGFBP7 [43] were associated with lower survival, whereas hypomethylation of CTCFL predicted higher tumor recurrence and lower survival [44]. A study including 1095 HCC and 835 controls generated a classifier of 8 markers for diagnosis and prognosis (*SH3PXD2A*, *C11orf9*, *PPFIA1*, *Chr 17:78*, *SERPINB5*, *NOTCH3*, *GRHL2* and *TMEM8B*). In addition to providing a high diagnostic accuracy, the score was also associated with survival [40]. Two other studies used comparable approaches and provided similar findings [39,41].

### 1.2. Circulating Free RNAs (cfRNAs)

RNAs include a large family of members: micro, long non-coding, messenger or exosomal RNAs. Herein, the present review will focus on the most commonly investigated circulating RNAs in liquid biopsies (Table 2).

#### 1.2.1. Micro-RNAs (miRNAs)

MiRNAs have gained increased interest as cancer biomarkers, as they have key properties, in particular their molecular stability. Lin et al. established a classifier based on seven miRNAs, which was able to detect preclinical HCC [67]. This could be a valuable tool for HCC surveillance, which could outperform the recommended bi-annual ultrasound (US) and AFP measurement.

A number of circulating miRNAs have shown prognostic value in HCC. Low levels of miR-1, miR-122, miR-26a, miR-29a and miR-223-3p were associated with lower survival [50,51,52,56]. Patients with high levels of miR-155, miR-96 and miR-193-5p had lower survival rates [53,55]. A recent study reported whole miRNome profiling in 116 HCC patients [54]. Using three different cohorts, the study reported on miRNAs differentially expressed in HCC vs. non-HCC patients. This effort identified miRNAs with specific clinical utilities; certain biomarkers detected cirrhosis, while others detected HCC. Furthermore, six miRNAs were identified as prognostic factors. Down-regulation of miR-424-5p or miR-101-3p and up-regulated miR-128, miR-139-5p, miR-382-5p and miR410 were associated with lower survival.

#### 1.2.2. Messenger RNAs (mRNAs)

Unlike miRNAs, circulating mRNAs are highly unstable and are thus rarely explored as liquid biopsy analytes. Studies attempting to measure mRNAs in blood samples analyzed a limited number of candidates.

A comparison of 50 HCC patients and 50 controls detected an association between VEGF expression level (isoform 165) and the risk of tumor recurrence [57]. The concentration of circulating mRNAs coding for AFP was investigated in two other studies. The first one included 38 HCC patients undergoing partial resection and showed that detection of AFP mRNA was associated with extrahepatic recurrence and shorter disease-free survival [58]. The second one tested both levels of AFP and hTERT mRNAs, but failed to identify any prognostic impact [59].

### 1.3. Extracellular Vesicles (EVs): Exosomes

Exosomes are a type of EV, nanoparticles encapsulating a variety of cargo including DNA and RNA fragments in a lipidic double-layer, which protects them from enzymatic degradation. With these unique features, circulating exosomes allow RNAs to circulate without being degraded plasma. Their nature and roles remain largely unknown but exosomes may not only be passively released from apoptotic cells into the bloodstream. Data suggested they may be actively secreted, acting as messengers in the cell-to-cell communication network, conferring them priceless values like accuracy and tissue-specificity [68,69,70,71].

In HCC, the data exploring the contribution of exosomes remain limited, particularly for prognosis. However, these analytes have demonstrated promising and polyvalent performance in other cancer types both for diagnosis and prognosis [72,73].

Several projects analyzed exosomal miRNAs. In a cohort of 59 HCC patients, authors found a correlation between tumor recurrence after liver transplantation and a higher level of miR-718 [60]. Similar signals were detected after liver resection and other exosomal miRNAs: high levels of miR-665 or low levels of miR-638 and miR-320a were identified as predictors of poor survival [61,63,66]. In a cohort of 79 HCC patients of different stages receiving various treatments, Lee et al. focused on two candidates: a miRNA (miR-21) and a long non-coding RNA (lncRNA) (lncRNA-ATB). On multivariable analysis, both markers were independently associated with disease progression [62]. A recent study profiled 57 plasma cell-free RNA transcriptomes and 20 exosomal RNA transcriptomes to test their diagnostic and prognostic performance. RN7SL1 and its S fragment were promising, showing a high diagnostic accuracy (AUC = 0.87). Furthermore, a higher concentration of RN7SL1 S fragment was an independent factor of worse survival [64]. The analyses of blood samples from 124 HCC patients treated with surgical resection and 100 healthy controls identified an exosomal circular RNA (circAKT3) as a prognostic factor; a high level of circAKT3 predicted both higher recurrence and lower survival [65].

### 1.4. Circulating Tumor Cells (CTCs)

CTCs play a pivotal role during the hematogenous dissemination of cancers. Most technologies to analyze CTCs include two steps: enrichment (isolation) and detection (identification). The development of sensitive and specific technologies is challenging. Estimated to be released by cancers of intermediate and advanced stages, CTCs are probably more useful for prognostication than for early cancer detection. In this context, studies have demonstrated the prognostic value of CTC enumeration in different cancers including HCC [74,75]. More sophisticated technologies, like single-cell RNA sequencing, have allowed further characterization of CTC subtypes [76]. Studies exploring the value of CTCs for HCC prognostication are summarized in Table 3.

Most studies investigating CTCs included HCC patients undergoing surgical resection. In 2004, Vona G et al. isolated and enumerated CTCs based on their size and morphology, showing that the presence and number of detected CTCs were associated with shorter survival [77]. CellSearch^®^ is an isolation system that targets EpCAM positive cells. It was approved by the Food and Drug Administration (FDA) and became the most commonly used technique for CTC enumeration. Its use remains debated in HCC as only around 30% of HCC cells express EpCAM [102]. Nonetheless, several studies utilized this approach in HCC, demonstrating that the detection of EpCAM positive CTCs was associated with higher tumor recurrence [75] or lower survival [82,87]. Thereafter, more sophisticated technologies for CTC isolation have been reported, like ImageStream flow cytometry. A study has provided the proof-of-concept of this technology, demonstrating its capacity to detect CTCs using a panel of markers. This technology also generates high-resolution images of isolated CTCs [103]. Its value in detecting CTCs was confirmed and the CTC count was further confirmed as an independent prognostic factor [86]. Other reports have aimed at exploring the impact of subgroups of CTCs, clustered based on cell surface markers, RNA expression or genomic aberrations. Studies using surface markers to detect CTCs with cancer stem cell-like [79,80,81] or mesenchymal [92] features, revealed their clinical value to predict tumor recurrence. CTCs expressing AFP were also associated with an increased risk of metastasis [85], whereas CTCs harboring CNV (chr 8) predicted worse survival [88]. Ha et al. used a simple isolation technique but introduced the concept of ΔCTC, referring to the perioperative fluctuation of detected CTCs, which appeared as an independent factor of lower survival and higher recurrence rates after partial hepatectomy [95]. Besides their intrinsic biological traits, CTC dissemination seemed to be impacted by treatment. Data suggested that surgery-induced manipulation of the liver is associated with a release of CTCs [94]. A comparison between anterior and conventional surgical approaches suggested that the latter was associated with a higher release of CTCs as well as poorer outcomes [89]. Toso et al. described five steps during orthotopic liver transplant (OLT) in HCC patients to minimize CTC dissemination and thereby the risk of recurrence [104].

Recent studies also underscored the relevance of CTC analysis in patients undergoing OLT for HCC, highlighting an association between CTC detection and recurrence [97,101].

Guiding decision-making would be another application of CTCs. For example, selection of patients who would benefit from adjuvant transarterial chemoembolization (TACE) after surgery or to predict response to systemic therapies like tyrosine kinase inhibitors or immunotherapy [84,99,100]. An interesting study analyzed CTCs using samples collected from different vessels. By doing so, they were able to demonstrate spatial heterogeneity in the distribution of CTCs, with a predominance of epithelial status at release, which gradually switched to EMT-activated phenotype during hematogenous transit [90]. Overall, data consistently identified that the number of CTCs was a surrogate of poor prognosis, predicting higher recurrence and/or lower survival. A recent meta-analysis and data from experimental models corroborated these findings [105,106].

## 2. Challenges and Future Perspectives

The field of liquid biopsy has numerous challenges. Besides those related to cost and technology, there is a limited understanding of the fundamental mechanisms responsible for the release of tumor molecular components to the bloodstream. A better understanding of these mechanisms would provide new tools and targets to improve the diagnostic and prognostic performance of liquid biopsy analytes. Gasparello et al. performed one of the few studies of liquid biopsies in animal models, identifying potential gateways regulating the detection of ctDNA [107]. Experimental models will be instrumental to better understand these mechanisms. ITH has emerged as a major drawback for single-biopsy biomarker development. The clinical impact of ITH is progressively recognized, even at early tumor stages [108]. Liquid biopsy can help address the clinical issues posed by ITH as it likely includes a molecular composite of tumor components released by any potential tumor area. Thus, it is not restricted by the specific tumor section sampled by a needle-biopsy. There are few data of integrative analysis of different analytes within the liquid biopsy space (e.g., simultaneous evaluation of ctDNA and CTCs). Finally, it is key to have prospective data to determine the exact role of liquid biopsy as a prognostic biomarker in HCC, and which is the clinical niche that will be better suited for this transformative technology.

## 3. Conclusions

While data on liquid biopsy in HCC remain scanter than for other malignancies, there has been numerous recent publications demonstrating its prognostic value in HCC patients. Potential contributions in HCC prognostication were detected for each of the tumor by-products (e.g., DNA, RNA, exosomes and cells). The next step will be to determine the optimal way of integrating liquid biopsy in the clinical management of HCC patients and to modify current clinical practice guidelines accordingly.

## Figures and Tables

**Figure 1 cancers-13-00659-f001:**
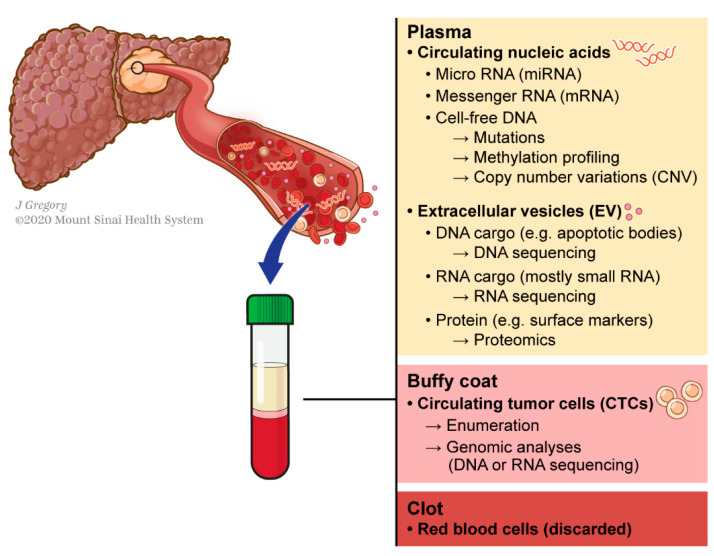
Concept of liquid biopsy referring to the molecular analysis of tumor-byproducts released into the bloodstream.

**Figure 2 cancers-13-00659-f002:**
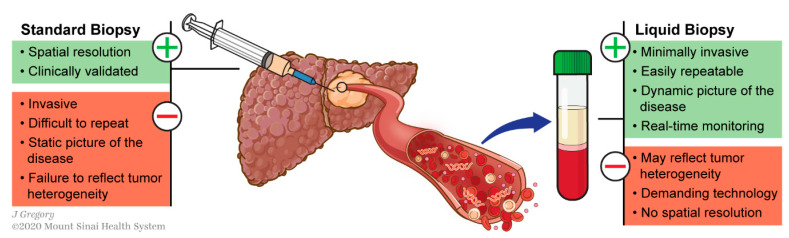
Pros and cons of liquid biopsy versus standard biopsy.

**Table 1 cancers-13-00659-t001:** Circulating tumor DNA (ctDNA).

Number of Patients	Treatment	Biomarkers	Technique	Main Finding	[Ref.]
**CNV**					
34 HCC	Surgery	ctDNA (harboring SNV or CNV)	Targeted-sequencing and low coverage whole-genome sequencing	ctDNA can detect minimal residual disease (MRD) and predict survival	[28]
151 HCC; 14 healthy controls	Sorafenib	*VEGFA* amplification	Whole-genome sequencing	High concentration of cell-free DNA (cfDNA) was associated with poor outcomes but VEGFA ratio was not a prognostic factor.	[29]
**Mutations**					
46 HCC	Surgery Transplant	ctDNA	Targeted-sequencing and exome-sequencing	Detection of ctDNA was associated with increased recurrence	[30]
41 HCC; 10 controls	Surgery	TERT, TP53 and CTNNB1	Targeted-sequencing	Detection of ctDNA predicted shorter recurrence-free survival	[31]
10 HCC	Surgery TACE RFA	Methylation of GSTP1 and RASSF1A or TP53 mutation	Methylation-specific PCR and sanger sequencing	Detecting ctDNA in urine was feasible and predicted recurrence	[32]
218 HCC; 81 cirrhotic	NA	*TERT* promoter mutation (C228T and C250T)	Droplet digital PCR (ddPCR) and sanger sequencing	TERT promoter mutation can be used as an early biomarker of HCC and is associated with survival	[33]
34 HCC	Surgery	ctDNA (harboring SNV or CNV)	Targeted-sequencing and low coverage whole-genome sequencing	ctDNA can detect minimal residual disease (MRD) and predict survival	[28]
95 HCC; 45 cirrhotic	Surgery	*TERT* promoter mutation (C228T)	Droplet digital PCR (ddPCR)	Detection of mutated *TERT* promoter was associated with lower survival	[34]
59 HCC	Surgery TACE RFA Systemic chemotherapy BSC	Single nucleotide variant (SNV) in a panel of 69 genes	Targeted-sequencing	Mutated MLH1 in plasma was associated with lower survival	[35]
130 HCC	TACE Systemic chemotherapy	*TERT* promoter mutation	Droplet digital PCR (ddPCR)	Detection of mutated *TERT* promoter was associated with lower survival	[36]
895 HCC	Surgery/NA	*TP53* mutation (R249S)	Droplet digital PCR (ddPCR)	Detection of mutated *TP53* was associated with lower survival	[37]
22 HCC	TKI (tyrosine kinase inhibitors)	Genes of the PI3K/MTOR pathway	Targeted-sequencing and ddPCR	Mutations of genes in the PI3K/MTOR pathway are associated with lower survival in patients treated with TKI	[38]
**Methylation Changes**					
72 HCC; 37 benign liver diseases; 41 healthy controls	-	APC, GSTP1, RASSF1A, and SFRP1	Methylation-specific PCR	Methylation of RASSF1A was associated with poor survival	[39]
1098 HCC; 835 controls	NA	8-marker panel	Targeted bisulfite sequencing	Methylation-based classifier predicted survival	[40]
10 HCC	TACE RFA Surgery	Methylation of GSTP1 and RASSF1A or TP53 mutation	Methylation-specific PCR and sanger sequencing	Detecting ctDNA in urine was feasible and predicted recurrence	[32]
203 HCC; 104 chronic viral hepatitis B or C; 50 healthy controls	NA	APC, COX2, RASSF1A (+miR-203)	Methylation-specific PCR	Classifier predicted survival	[41]
172 HCC	NA	LINE-1	Methylation-specific PCR	Hypomethylation of LINE-1 was associated with lower survival	[42]
155 HCC; 60 chronic HBV; 20 healthy controls	Surgery	IGFBP7	Methylation-specific PCR	Methylation of IGFBP7 was associated with lower survival	[43]
43 HCC (+347 HCC from TCGA Atlas); 5 cirrhotic; 6 benign liver lesions	-	CTCFL	Methylation-specific PCR	Hypomethylation of CTCFL was associated with higher recurrence and lower survival	[44]

**Table 2 cancers-13-00659-t002:** Circulating free RNAs (cfRNAs) and exosomes.

Number of Patients	Treatment	Biomarkers	Technique	Main Finding	[Ref.]
**miRNA**					
195 HCC 54 cirrhotic	Surgery Transplant TACE RFA sorafenib	miR-1 and miR-12	qRT-PCR	Low level of miR-1 was associated with lower survival	[50]
122 HCC	Surgery	miR-122	qRT-PCR	Low level of miR-122 was associated with lower survival	[51]
120 HCC	Surgery RFA	MiR-21, miR-26a, and miR-29a	qRT-PCR	Low levels of miR-26a and miR-29a were associated with lower survival	[52]
30 HCC; 30 controls	Surgery	miR-155, miR-96 and miR-99a	qRT-PCR	High levels of miR-155 and miR-96 were associated with lower survival	[53]
116 HCC	NA	Circulating miR	Whole miRNome profling	Low levels of miR-424-5p, miR-101-3p or high levels of miR-128, miR-139-5p, miR-382-5p and miR410 were associated with lower survival	[54]
41 HCC; 20 controls	Surgery transplant	miR193a-5p	qRT-PCR	High level of miR193a-5p was associated with lower survival	[55]
70 HBV-related HCC 70 HBV 50 healthy controls	Surgery	miRNA-223-3p	qRT-PCR	Low level of miRNA-223-3p was associated with lower survival	[56]
**mRNA**					
50 HCC; 50 controls	Surgery	VEGF-165	qRT-PCR	Detection of circulating VEGF mRNA (isoform 165) was associated with higher recurrence and recurrence-related mortality	[57]
38 HCC	Surgery	AFP	qRT-PCR	Detection of AFP mRNA was associated with extrahepatic recurrence and shorter disease-free survival	[58]
343 HCC	Surgery TACE RFA Systemic chemotherapy Radiotherapy BSC	AFP and hTERT	qRT-PCR	Detection of AFP mRNA or hTERT mRNA was not associated with survival	[59]
**Exosomes**					
59 HCC	Transplant	miR-718	qRT-PCR	Recurrence was associated with higher level of exosomal miR-718	[60]
30 HCC	Surgery	miR-665	qRT-PCR	High level of exosomal miR-665 was associated with lower survival	[61]
79 HCC	Surgery Transplant TACE RFA Sorafenib BSC	miR-21 and lncRNA-ATB	qRT-PCR	High levels of exosomal miR-21 and lncRNA-ATB were associated with lower survival	[62]
126 HCC; 21 healthy controls	Surgery	miR-638	qRT-PCR	Low level of exosomal miR-638 was associated with lower survival	[63]
31 HCC; 3 CLD; 11 healthy controls	NA	RN7SL1 S fragment	qRT-PCR	High expression of RN7SL1 S fragment was associated with lower survival	[64]
124 HCC; 100 healthy controls	Surgery	AKT3	qRT-PCR	High level of exosomal circulating AKT3 was associated with higher recurrence and lower survival rates	[65]
104 HCC; 55 CLD; 50 healthy controls	Surgery	miR-320a	qRT-PCR	Low serum exosomal miR-320a was associated with lower survival	[66]

**Table 3 cancers-13-00659-t003:** Circulating tumor cells (CTCs).

Number of Patients	Treatment	Technique of Detection	Main Finding	[Ref.]
44 HCC 30 HCV 39 cirrhosis 38 healthy controls	Surgery NA	Isolation by size of epithelial tumor cells (ISET)	Presence and number of detected CTCs were associated with shorter survival	[77]
85 HCC 37 benign liver diseases 20 healthy volunteers 14 miscellaneous advanced cancers other than HCC	Surgery NA	Antibody-coated magnetic beads	Presence and number of detected CTCs correlated with tumor size, portal vein tumor thrombus, differentiation status, TNM stage and Milan criteria	[78]
82 HCC	Surgery	Multicolor flow cytometry	Circulating cancer stem cells (CSC) are associated with higher rates of intra- and extra-hepatic recurrence, decreased recurrence-free survival (RFS) and overall survival (OS) rates	[79]
96 HCC 31 healthy controls 21 viral hepatitis 8 cirrhosis	Surgery	Magnetic cell sorting (Lin28B)	Detection of CTCs expressing Lin28B was associated with early recurrence	[80]
60 HCC	Surgery NA	Flow cytometry (ICAM-1)	Detection of CTCs expressing ICAM-1 was associated with shorter disease-free survival	[81]
123 HCC	Surgery	EpCAM antibody-coated magnetic beads (CellSearch)	Detection of CTCs (EpCAM+) was associated with higher recurrence	[75]
59 HCC 19 controls	NA	EpCAM antibody-coated magnetic beads (CellSearch)	Detection of CTCs was associated with lower overall survival	[82]
122 HCC 120 controls	Surgery TACE Radiotherapy	EpCAM antibody-coated magnetic beads (CellSearch)	Peri-treatment decrease of detected CTC reflected treatment response	[83]
109 HCC	Surgery TACE RFA sorafenib	Flow cytometry (ASGPR and CPS1)	pERK+/pAkt-CTCs correlated with progression-free survival and predicted response to systemic therapy (sorafenib)	[84]
72 HCC	Surgery	EpCAM antibody-coated magnetic nanoparticals (MagVigen, Nvigen)	Detection of CTCs expressing AFP was associated with metastatic disease	[85]
69 HCC 31 controls	Surgery Transplant TACE RFA Sorafenib BSC	Imaging flow cytometry (EpCAM, AFP, glypican-3 and DNA-PK together with analysis of size, morphology and DNA content) (ImageStream)	Detection of CTCs was associated with lower survival	[86]
57 HCC	Surgery	EpCAM antibody-coated magnetic beads (CellSearch)	CTCs detection was associated with higher recurrence and lower recurrence-free survival after liver resection	[87]
14 HCC 16 CCA 4 GBC	Surgery	SE-iFISH	Detection of small CTCs with CNV (chromosome 8) was associated with lower survival	[88]
199 HCC	Surgery	Fluorescence-activated cell sorting (FACS)	Anterior approach was associated with a decreased dissemination of CTCs compared to conventional approach, resulting in poorer outcomes.	[89]
73 HCC	Surgery	EpCAM antibody-coated magnetic beads (CellSearch)	Analyzes of blood samples collected in different vessels revealed a spatial heterogeneity of CTCs distribution whose biology was associated with recurrence pattern.	[90]
130 HCC	Surgery TACE	qRT-PCR test platform	CTCs detection was associated with recurrence after liver resection	[91]
112 HCC	Surgery	CanPatrolTM system (filtration by size) and Tri-color RNA-ISH assay	The presence of CTCs and the proportion of mesenchymal-CTC (M-CTCs) were associated with recurrence	[92]
61 HCC 19 non-HCC	TACE TARE RFA Systemic therapy	Antibody-based platform	Vimentin (VIM)-positive CTCs predicted OS and faster recurrence after curative-intent surgical or locoregional therapy in potentially curable early-stage HCC	[93]
139 HCC 23 controls	Surgery	EpCAM antibody-coated magnetic beads (CellSearch)	Surgical resection induces a release of CTCs	[94]
105 HCC	Surgery	ISET	ΔCTCs is an independent predictor of lower survival and higher recurrence in patients	[95]
85 HCC 27 non-HCC	Surgery	Flow cytometry (GPC3)	GPC3 positive-CTCs detection was associated with lower survival	[96]
50 HCC	Transplant	Negative enrichment and immunofluorescence in situ hybridization (imFISH)	CTCs detection was associated with early recurrence after liver transplant	[97]
137 HCC	Surgery	ISET	CTCs detection was associated with early recurrence after liver resection	[98]
87 HCC 7 cirrhosis 8 healthy controls	Transplant Surgery TACE TARE RFA Systemic therapy	Antibody-based platform	Detection of CTCs expressing PD-L1 were associated with shorter OS and predicted response to immunotherapy	[99]
128 HCC	Surgery ± TACE	EpCAM antibody-coated magnetic beads (CellSearch)	Adjuvant TACE provided survival and recurrence benefits in patients with positive preoperative CTCs	[100]
193 HCC	Transplant	Antibody-based platform (ChimeraX®-i120)	CTCs detection was associated with recurrence after liver transplant	[101]

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
