# Peer review of "The Role of Liquid Biopsy in Hepatocellular Carcinoma Prognostication"

_cancers, 2021, doi:10.3390/cancers13040659_

Round 1
Reviewer 1 Report
In the present manuscript, Lagbaa et al. nicely summarize the available data and the relevance of liquid biopsy in human hepatocellular carcinoma, an aggressive liver tumor with limited therapeutic options. The review article is comprehensive, well-written, logically structured, and highly relevant for basic and clinical liver cancer research. The appropriate references are cited, and the figures are easy to understand as well.
Minor issue:
- A table summarizing the major pros and cons of the various strategies for HCC prognostication would be highly helpful for the readers.
Author Response
In order to assist reviewers, we have established 3 writing modes in this document: black-bold for the original reviewer’s comment as included in the decision letter, black-regular for our responses to reviewer’s comments and red for the changes made to the original version of the manuscript.
REVIEWER 1
In the present manuscript, Lagbaa et al. nicely summarize the available data and the relevance of liquid biopsy in human hepatocellular carcinoma, an aggressive liver tumor with limited therapeutic options. The review article is comprehensive, well-written, logically structured, and highly relevant for basic and clinical liver cancer research. The appropriate references are cited, and the figures are easy to understand as well.
We would like to thank the reviewer for his/her very positive comments and for highlighting that this review was comprehensive and that the appropriate references were cited.
Minor issue:
A table summarizing the major pros and cons of the various strategies for HCC prognostication would be highly helpful for the readers.
We understand this pertinent comment. As this review will be part of a special and dedicated issue on the prognosis in HCC, we guess that other articles of the issue will address other HCC prognostication systems. Therefore, our review is probably not the most appropriate place for a table providing the pros and cons of these various systems.
However, we felt that adding a figure (Figure 2) providing the pros and cons of liquid biopsy vs. standard biopsy for HCC prognostication may be valuable for the readers, also somehow addressing the reviewer’s comment.
Reviewer 2 Report
The review indicates recent progress and possible prospects for the use of liquid biopsy in HCC. The manuscript is overall well written and offers an important point of reflection on the need to direct research efforts in HCC towards the development of more performing liquid biopsy technologies capable of allowing complete patient monitoring, combining the various approaches of liquid biopsy and thus providing a complete picture of the circulating state of the disease.
However, although the review is very valid from a conceptual point of view, there is a need to expand the general information that appears to be a bit poor in content.
1- Line 49-55 is to expand, better explaining the liquid biopsy and the concept of the tumor circulome (PMID: 30736982)
2- "microRNA and exosomes" it should not be a single paragraph and exosomes deserve to have more importance since they are the clear mirror of the "tumor burden" as they are produced in large quantities in cancer patients compared to healthy controls, perhaps indicating their responsibility in tumor communication. Again, reference should be made to exosomal diversity because not only in liquid biopsy it is essential to understand which is the right approach, but within every single approach, it is necessary to take into account the diversity of the individual populations of the circuloma. In particular, in the case of exosomes, it is essential to be able to obtain a unique biomarker (PMID: 32759810)
3 - paragraph 1.2.3: At this point, it is poor in references, I suggest integrating and constituting a more substantial paragraph (PMID 29029605; PMID: 32759810; PMID: 32937811) such as the paragraph on CTCs which is very valid.
With these suggestions, the authors may extend the review, considering the normal length of this type's manuscripts and updating a good part of references.
Finally, there are some minor problems of English spelling check
line 45 - a relatively
line 174 - as a prognostic
line 209- surgery-induced
line 224 - It appears that an article is missing before the word number. Consider adding the article.
About tables S1 and S2: I don't think there is a need to enter the authors and the journal's names.
Table S2: I suggest to change the PMID column title to Ref, as in table 1.
Author Response
In order to assist reviewers, we have established 3 writing modes in this document: black-bold for the original reviewer’s comment as included in the decision letter, black-regular for our responses to reviewer’s comments and red for the changes made to the original version of the manuscript.
REVIEWER 2
The review indicates recent progress and possible prospects for the use of liquid biopsy in HCC. The manuscript is overall well written and offers an important point of reflection on the need to direct research efforts in HCC towards the development of more performing liquid biopsy technologies capable of allowing complete patient monitoring, combining the various approaches of liquid biopsy and thus providing a complete picture of the circulating state of the disease.
We thank the reviewer for his/her encouraging comment and for underscoring the relevance of this manuscript.
However, although the review is very valid from a conceptual point of view, there is a need to expand the general information that appears to be a bit poor in content.
- Line 49-55 is to expand, better explaining the liquid biopsy and the concept of the tumor circulome (PMID: 30736982)
The reviewer raised a good point; we further elaborated this paragraph and added a figure (Figure 2).
Page 2 (lines 55-61)
This technology offers a valuable alternative to standard biopsy. Tissue biopsy is indeed invasive and associated with potential risks such as pain, bleeding or even seeding of the cancer (PMID: 18669577). Conversely to standard biopsy, liquid biopsy displays the advantages of being easily repeatable and can thereby help for monitoring, providing a dynamic picture of the disease course. In addition, it may reflect different regions of the tumor and thus recapitulate eventual intra-tumoral heterogeneity (ITH) (Figure 2) (PMID: 30736982).
- "microRNA and exosomes" it should not be a single paragraph and exosomes deserve to have more importance since they are the clear mirror of the "tumor burden" as they are produced in large quantities in cancer patients compared to healthy controls, perhaps indicating their responsibility in tumor communication. Again, reference should be made to exosomal diversity because not only in liquid biopsy it is essential to understand which is the right approach, but within every single approach, it is necessary to take into account the diversity of the individual populations of the circuloma. In particular, in the case of exosomes, it is essential to be able to obtain a unique biomarker (PMID: 32759810)
We followed the reviewer’s suggestion and separated paragraphs on Circulating free RNAs (cfRNAs) and exosomes; paragraphs on exosomes and CTCs thus became #1.3. and #1.4., respectively.
We also expanded on exosomes as not all readers are familiar with this entity:
Page 4 (lines 166-170)
Their nature and roles remain largely unknown but exosomes may not only be passively released from apoptotic cells into the bloodstream. Data suggested they may be actively secreted, acting as messengers in the cell-to-cell communication network, conferring them priceless values like accuracy and tissue-specificity 32759810 / 32795414 / 26524530 / 25985394.
- paragraph 1.2.3: At this point, it is poor in references, I suggest integrating and constituting a more substantial paragraph (PMID 29029605; PMID: 32759810; PMID: 32937811) such as the paragraph on CTCs which is very valid.
We understand the idea of further emphasizing exosomes relevance and therefore to further expand this paragraph. Nonetheless, this paragraph aims to review the data on exosomes for HCC prognostication, not to discuss exosomes in cancers. None of the references cited by the reviewer are related to HCC, but rather to other cancer types.
Nonetheless, we also agree that exosomes are particularly interesting analytes which deserve to be better explored in the future. Hence, we expanded this paragraph to provide more information, citing the suggested references.
Page 4 (lines 171-173)
In HCC, the data exploring the contribution of exosomes remain limited, particularly for prognosis. However, these analytes have demonstrated promising and polyvalent performance in other cancer types both for diagnosis and prognosis (PMID: 29029605, 32937811).
With these suggestions, the authors may extend the review, considering the normal length of this type's manuscripts and updating a good part of references.
Finally, there are some minor problems of English spelling check
line 45 - a relatively
We modified line 95 according to the reviewer’s comment (Page 3; line 101)
line 174 - as a prognostic
Modified according to the reviewer’s comment (Page 5; line 187)
line 209- surgery-induced
Modified according to the reviewer’s comment.
The manuscript now reads:
Data suggested that surgery-induced manipulation of the liver is associated with a release of CTCs (Page 5; line 222)
line 224 - It appears that an article is missing before the word number. Consider adding the article.
This sentence was modified, for clarification. The manuscript now reads:
Overall, data consistently identified that the number of CTCs was a surrogate of poor prognosis predicting higher recurrence and/or lower survival (Page 6; line 237).
About tables S1 and S2: I don't think there is a need to enter the authors and the journal's names.
Following the reviewer’s comment, we removed columns with authors and journals’ names.
Table S2: I suggest to change the PMID column title to Ref, as in table 1.
The column’s title “PMID” was changed for “Ref”, as suggested by the reviewer.
Round 2
Reviewer 2 Report
I thank the authors for their comprehensive replies to my comments. Very good the added figure 2 and the inserted passages, now it is a more consistent manuscript. Nothing else to add, well done!